# A Method for Detecting Atmospheric Lagrangian Coherent Structures Using a Single Fixed-Wing Unmanned Aircraft System

**DOI:** 10.3390/s19071607

**Published:** 2019-04-03

**Authors:** Peter J. Nolan, Hunter G. McClelland, Craig A. Woolsey, Shane D. Ross

**Affiliations:** 1Engineering Mechanics Program, Virginia Tech, Blacksburg, VA 24061, USA; sdross@vt.edu; 2Kevin T. Crofton Department of Aerospace and Ocean Engineering, Virginia Tech, Blacksburg, VA 24061, USA; hgm@vt.edu (H.G.M.); cwoolsey@vt.edu (C.A.W.)

**Keywords:** unmanned aircraft system (UAS), Lagrangian coherent structure (LCS), atmospheric transport

## Abstract

The transport of material through the atmosphere is an issue with wide ranging implications for fields as diverse as agriculture, aviation, and human health. Due to the unsteady nature of the atmosphere, predicting how material will be transported via the Earth’s wind field is challenging. Lagrangian diagnostics, such as Lagrangian coherent structures (LCSs), have been used to discover the most significant regions of material collection or dispersion. However, Lagrangian diagnostics can be time-consuming to calculate and often rely on weather forecasts that may not be completely accurate. Recently, Eulerian diagnostics have been developed which can provide indications of LCS and have computational advantages over their Lagrangian counterparts. In this paper, a methodology is developed for estimating local Eulerian diagnostics from wind velocity data measured by a single fixed-wing unmanned aircraft system (UAS) flying in a circular arc. Using a simulation environment, driven by realistic atmospheric velocity data from the North American Mesoscale (NAM) model, it is shown that the Eulerian diagnostic estimates from UAS measurements approximate the true local Eulerian diagnostics and also predict the passage of LCSs. This methodology requires only a single flying UAS, making it easier and more affordable to implement in the field than existing alternatives, such as multiple UASs and Dopler LiDAR measurements. Our method is general enough to be applied to calculate the gradient of any scalar field.

## 1. Introduction

The transport of material in the atmosphere is a problem with important implications for agriculture [1,2,3,4], aviation [5,6], and human health [7,8]. Given the unsteady nature of atmospheric flows, it can be difficult to predict where a fluid parcel, such as one containing an airborne pathogen, will be transported. Tools from dynamical systems theory, such as Lagrangian coherent structures (LCSs), can help us to understand how fluid parcels in a flow will evolve. The study of atmospheric transport from a dynamical systems perspective has long focused on the study of large-scale phenomena [1,2,3,4,5,9,10,11,12]. This has been largely due to the larger-scale grid spacing of readily available atmospheric model data and the lack of high-resolution atmospheric measurements on a scale large enough to calculate Lagrangian data. Furthermore, this field of study has been largely relegated to numerical simulations. Few works have attempted to find ways to directly detect LCSs from experimental measurements in the field. In [6,13], the authors used wind velocity measurements from a Doppler light detection and ranging (LiDAR) to detect LCS which had passed near Hong Kong International Airport. Rather than measure the wind velocity to try to detect LCSs, the authors of [1] looked at sudden changes in pathogen concentrations in the atmosphere. They were then able to link those changes to the passage of LCSs using atmospheric velocity data from the North American Mesoscale (NAM) weather research and forecasting (WRF) model. Recent advances in dynamical systems theory, such as new Eulerian diagnostics, as well as new atmospheric sensing technology, such as unmanned aircraft systems (UAS) [14], have brought the local detection of LCSs within the reach of operators in the field at any location, without the need for expensive infrastructure. One recent paper took advantage of these new advances to attempt to predict the passage of LCSs based on experimental data. The authors of [15] used multiple sonic anemometers attached to two stationary airborne quadcopter UASs and one ground-based tower to measure the wind velocity around the San Luis Valley in Colorado. From these wind measurements, the authors were able to calculate the local attraction rate, which was then compared to the attraction rate from a WRF simulation. From these comparisons, the authors found indicators of LCSs associated with convective cells and a front which passed through the region. In this paper, we build upon these recent developments to establish a method to detect LCSs which can be readily implemented by operators in the field at any location, using only a single fixed-wing UAS.

The first of these developments is new Eulerian techniques for measuring the attraction and repulsion of regions in a fluid flow [16,17]. In traditional Lagrangian analyses, a velocity field is needed, which is defined over a large enough spatiotemporal scale to allow the accurate simulation of fluid parcel trajectories. How the parcels are transported by the flow is then used to determine which parts of the flow are more attractive or repulsive. These new Eulerian methods do not rely on the simulation of fluid parcel trajectories; instead, they are based on the instantaneous gradients of the velocity field. Since they rely on gradients, these techniques only require enough velocity data points to enact a finite-differencing scheme. Furthermore, these methods are Eulerian and can thus be applied to data sets which are temporally coarse.

The second of these developments is the use of inexpensive UASs to sample the atmospheric velocity instead of piloted aircraft or other traditional assets. Ground-based wind sensors such as LiDAR, sonic detection and ranging (SoDAR), or tower-mounted anemometers can be prohibitively expensive and difficult to relocate in real time to regions of interest, such as a chemical release, wild fire, or radioactive release. Airborne wind measurement from aircraft has a long history [18,19] and well-developed existing programs, such as the NASA Airborne Science Program [20]. The advancement of UASs has enabled wind measurement missions which may be of lower cost, longer duration, and can be implemented in more dangerous environments. Elston et al. [21] provided a review of many UAS atmospheric measurement efforts, and recent works continue to advance both theoretical and practical UAS capabilities [22,23,24,25,26,27].

This paper develops a methodology which will enable researchers in the field to utilize local wind measurements to detect LCSs using only a single UAS. This methodology will be less complex, less expensive, and less time-consuming to implement than those utilized in previous studies [1,6,13,15]. Furthermore, by requiring only a single fixed-wing UAS, the need for expensive infrastructure and multiple pilot/spotter teams is eliminated. Additionally, since this methodology takes advantage of Eulerian diagnostics, there is no need for expensive computer clusters or time-consuming trajectory integration, as with Lagrangian diagnostics. This methodology is developed and tested using an observing system simulation experiment (OSSE). A historical wind velocity forecast from the NAM model is taken and passed to a UAS flight simulator, which is assumed to have a perfect anemometer, but still includes the effects of the wind field on the aircraft’s flight trajectory. The simulated UAS attempts to fly in a circular orbit about a portion of the NAM model centered on the coordinates of the Virginia Tech experimental site, Kentland Farm, Figure 1. From wind measurements along these circular orbits, we were able to calculate the local trajectory divergence rate and attraction rate, described in Section 2.1. Using these calculations, it is then possible to look for signals of LCSs passing through the area.

This paper is organized as follows. Section 2.1 describes the Eulerian and Lagrangian diagnostics which are used to analyze the atmospheric velocity data. Section 2.2 describes the algorithm we develop to calculate the gradient of a scalar field from measurements along a circular arc, which is necessary for the computation of the Eulerian diagnostics. Section 2.3 briefly describes the NAM model which is used, how the model data are prepared for analysis, and the flight simulator which is used. In Section 3.1, the results of a parametric study which analyzes the ability of a UAS flying in a circular arc to approximate Eulerian diagnostics over various radii are presented. Section 3.2 presents results concerning the ability of Eulerian diagnostics to approximate Lagrangian diagnostics and detect LCSs passing through an area using receiver operating characteristic (ROC) curves. Due to the discrete nature of numerical computations, a parametric study to investigate various area thresholds for the detection of LCSs was performed. Section 3.3 presents results concerning the ability of Eulerian diagnostics, as approximated from a simulated UAS flight, to detect LCSs passing through an area. Once again, a parametric study to investigate various area thresholds for the detection of LCSs using ROC curves is performed. Section 4 discusses the results presented in Section 3. Finally, Section 5 presents the conclusion of the paper.

## 2. Methods

### 2.1. Lagrangian-Eulerian Analysis

Consider the dynamical system:(1)$$\begin{array}{cc} \frac{d}{dt}x\left(t\right) & =v(x(t),t), \end{array}$$
(2)$$\begin{array}{cc} x_{0} & =x(t_{0}), \end{array}$$
(3)$$\begin{array}{cc} x(t)\in & R^{2},t\in R, \end{array}$$
where x(t) is the position vector of a fluid parcel at time *t* and v(x,t) is the time-varying horizontal wind velocity vector at position x(t) and time *t*. The components of the horizontal position vector are x=(x,y), where *x* is the eastward position and *y* is the northward position, and the horizontal velocity vector, v=(u,v), where *u* is the eastward velocity and *v* is the northward velocity. This system can be analyzed using both Langrangian and Eulerian tools. For the Lagrangian analysis, the finite-time Lyapunov exponent (FTLE), σ, and Lagrangian coherent structures (LCSs) are used. For this study, LCSs are defined as C-ridges of the FTLE field following [28]. The FTLE field is a measure of the stretching of fluid parcels within a flow, the forward-time FTLE measures repulsion, and the backward-time FTLE measures attraction. LCSs are the most attracting and repelling material surfaces within a fluid flow; as such, they provide a means of visualizing how particles within the flow will evolve. For the Eulerian analyses, the attraction rate, $s_{1}$, and the trajectory divergence rate, $\dot{\rho }$, are used. Both of these rates are derived from the Eulerian rate-of-strain tensor, S, described below. The attraction rate is the minimum eigenvalue of S and was shown in [16] to provide a measure of instantaneous hyperbolic attraction, with isolated minima of $s_{1}$ providing the cores of attracting objective Eulerian coherent structures (OECS). Recent work has shown that in 2D, $s_{1}$ is the limit of the backward-time FTLE as the integration time goes to 0 and that troughs of $s_{1}$ are attracting infinitesimal-time LCS (iLCS) [29], a schematic of which can be seen in Figure 2. The trajectory divergence rate is a measure of how much repulsion/attraction is changing along streamlines of the velocity field, as depicted in Figure 3 [17].

To calculate the Lagrangian metrics, first calculate the flow map of the vector field (Equation 1) for the time period of interest:(4)$$F_{t_{0}}^{t}\left(x_{0}\right)=x_{0}+\int_{t_{0}}^{t}v\left(x\left(t\right),t\right)\;dt.$$

Taking the gradient of the flow map, the right Cauchy–Green strain tensor is then calculated:(5)$$C_{t_{0}}^{t}\left(x_{0}\right)=\nabla F_{t_{0}}^{t}{\left(x_{0}\right)}^{T}\cdot \nabla F_{t_{0}}^{t}\left(x_{0}\right),$$
with ordered eigenvalues, $\lambda_{1}\leq \lambda_{2}$. From the largest eigenvalue of the right Cauchy–Green strain tensor, $\lambda_{2}$, the FTLE field is derived:(6)$$\sigma_{t_{0}}^{t}\left(x_{0}\right)=\frac{1}{2|t-t_{0}|}\log\left(\lambda_{2}\left(x_{0}\right)\right).$$

Ridges of this field can then be identified as LCSs.

For the Eulerian metrics, the Eulerian rate-of-strain tensor is defined as:(7)$$S\left(x_{0},t_{0}\right)=\frac{1}{2}\left(\nabla v\left(x_{0},t_{0}\right)+\nabla v{\left(x_{0},t_{0}\right)}^{T}\right),$$
with ordered eigenvalues $s_{1}\leq s_{2}$. The attraction rate is defined as the minimum eigenvalue of S, $s_{1}$. The attraction rate can be directly calculated from the velocity gradient using the formula:(8)$$s_{1}=\frac{1}{2}\left(\frac{\partial u}{\partial x}+\frac{\partial v}{\partial y}\right)-\frac{1}{2}\sqrt{{\left(\frac{\partial u}{\partial x}-\frac{\partial v}{\partial y}\right)}^{2}+{\left(\frac{\partial u}{\partial y}+\frac{\partial v}{\partial x}\right)}^{2}}.$$

The trajectory divergence rate is defined wherever $v(x_{0},t_{0})\neq 0$ as:(9)$$\dot{\rho }\left(x_{0},t_{0}\right)=\hat{n}{\left(x_{0},t_{0}\right)}^{T}\cdot S\left(x_{0},t_{0}\right)\cdot \hat{n}\left(x_{0},t_{0}\right)=\frac{\left(v{\left(x_{0},t_{0}\right)}^{T}\cdot J^{T}\cdot S\left(x_{0},t_{0}\right)\cdot J\cdot v\left(x_{0},t_{0}\right)\right)}{\|v\left(x_{0},t_{0}\right){\|}^{2}},$$
where $\hat{n}\left(x_{0},t_{0}\right)$ is the unit vector normal to the trajectory and J is the matrix, $J=\left[\begin{array}{cc} 0 & 1\\ -1 & 0 \end{array}\right]$ [17].

### 2.2. Gradient Approximation from UAS Flight

In order to calculate the Eulerian rate-of-strain tensor from wind measurements along a simulated UAS flight path, we developed an algorithm to approximate the gradient of a scalar field based on measurements along a circular arc, Figure 4. The algorithm is based on a quasi-frozen field assumption that the scalar field is not significantly changing in time during the period of one full orbit but is changing in space. We believe this assumption is appropriate to apply to atmospheric velocity fields, as mid- to larger-scale atmospheric flows tend to change on the order of hours, while UAS orbits on the spatial scale of interest are on the order of minutes. This assumption of course ignores small scale turbulent motion, which would fall below the scale which is being sampled. This algorithm also assumes that the important features will be in the horizontal plane. This assumption was previously applied to atmospheric model data in [9,10,30] and atmospheric measurements in [15]. It is based on the fact that the vertical component of the wind velocity tends to be two orders of magnitude less than the horizontal components. We assume that the inertial velocity of the aircraft, V, and the air-relative velocity, $V_{r}$, are independently measured, using, for instance, GPS-based instruments, orientation information from a gyroscope, and a pitot tube anemometer. The wind velocity is $v=V-V_{r}$. See Appendix A for further details.

We remark that while we developed the algorithm described below for measurements of the gradient of the horizontal wind velocity components, *u* and *v*, the algorithm is general enough to be applied to the gradient of any scalar value, *f*, measured by a UAS, such as air temperature, pressure, and humidity.

This algorithm takes as inputs:A scalar field measured along a circular arc, f(θ), as an n×1 array input, where *n* is the number of measurements taken;The angle θ as a monotonically decreasing n×1 array input;And the radius of the circular arc, *r*, which is assumed to be constant, as a scalar input.

Note, this algorithm is currently written for a clockwise trajectory. The algorithm starts with an initial point along the circular arc $\left(r,\theta_{0}\right)$ and the value *f* at that point. Then, provided the path continues for at least another three quarters of a circle, the value of *f* is obtained at three additional points along the path at $\left(r,\theta_{0}-\frac{1}{2}\pi \right),\left(r,\theta_{0}-\pi \right),\;\mathrm{and}\;\left(r,\theta_{0}-\frac{3}{2}\pi \right)$. With *f* at four individual points along the circular arc, a central finite-difference scheme is used to approximate the gradient of *f* at the center point of the circle. Since the four points are along an arc, each subsequent approximation the gradient will be in a reference frame which has a different orientation from the initial gradient approximation, Figure 4. To correct for this, apply a counterclockwise rotation to the gradient of *f* to obtain the gradient in a consistent reference frame oriented North–South, East–West. Continue for each additional point along the circular arc until there is less than an additional three quarters of a circle left. A pseudo-code version of this algorithm can be found in Algorithm 1, and a schematic can be found in Figure 4.

**Algorithm 1** Circle Gradient approximates the gradient of a scalar from samples along an arc

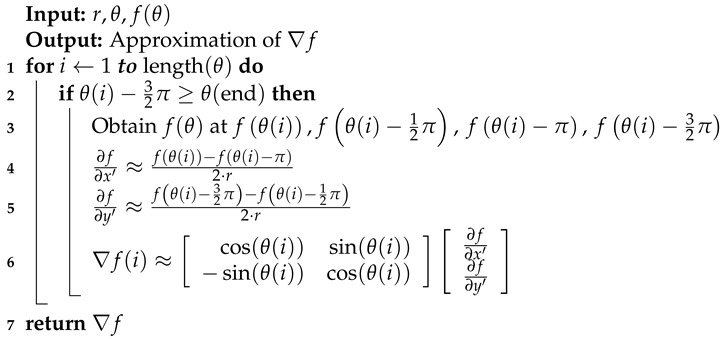



### 2.3. Model Data

For the OSSE, velocity field data from the 3 km NAM model [31] were utilized. This velocity field was from the portion of the model over southwestern Virginia and was centered at the Virginia Tech experimental site, Kentland Farm (Figure 1), during a 215 h period beginning 4 September 2017 at 00:00 UTC. The NAM data were divided into 2 data sets. The first part was a strictly 2D data set that was restricted to the 850 mb isosurface. The second was a 3D data set. Both data sets were interpolated in time from 1 h resolution down to 10 min resolution using cubic splines. The 3D data were then interpolated from pressure-based vertical levels to height above sea level (ASL) vertical levels using linear interpolation. Both data sets were also interpolated from a 3 km horizontal resolution to a 300 m horizontal resolution using cubic Lagrange polynomials. An aircraft was simulated using closed-loop trajectory control to fly fixed-radius circles. These circles varied in radius from 2 km to 15 km. Smaller radii, more consistent with the federal regulations in FAA part 107 (i.e., to keep aircraft within unaided sight) [32], were examined; however, the simulated wind velocity field, with a grid resolution of 3 km, lacked the spatial inhomogeneity necessary to compute meaningful gradients below a 2 km flight radius (4 km flight diameter). The simulated aircraft was also set to track the 850 mb isosurface with the 3D wind velocity field acting as a disturbance. The altitude of the 850 mb isosurface varies in time; it is approximately at 1545 m ASL. A subscale model of a transport-style aircraft, named the T-2 [33], was used as the simulated unmanned aircraft. To get a sense of the aircraft’s scale, some of its physical properties are:$$\mathrm{mass}\;m=22.5\;\mathrm{kg},\;\mathrm{wingspan}\;b=2.09\;m,\;\mathrm{chord}\;\bar{c}=0.28\;m,\;\mathrm{airspeed}\;\|V_{r}\|\approx 42\;m/s.$$

With the T-2’s cruising airspeed of approximately 42 m/s, a full orbit takes between 300 to 2250 s, depending on the orbit radius. For a comparison, the average horizontal wind speed of the NAM model at the 850 mb isosurface was approximately 10 m/s. The details of the flight dynamic model are included in Appendix A. The simulated wind measurements taken by the aircraft are wind field components v along the aircraft’s center-of-mass trajectory x(t). Figure 5 shows several orbits during the aircraft’s flight. Note that the vertical dimension is scaled to be 500 times bigger than the horizontal, in order to show flight path differences between consecutive orbits.

## 3. Results

### 3.1. Approximating Local Eulerian Metrics from UAS Flights

This section presents results which indicate how well the attraction rate, $s_{1}$, and the trajectory divergence rate, $\dot{\rho }$, can be approximated from a UAS flight. Figure 6 shows the results for the trajectory divergence rate. From the 2D velocity field, restricted to the 850 mb isosurface, measurements along perfectly circular paths, with radii fixed between 2 km and 15 km, were used to approximate the trajectory divergence rate, shown in black. Then, velocity measurements from 3D simulated UAS flight paths with radii fixed between 2 km and 15 km, attempting to follow the 850 mb isosurface, were used to approximate the trajectory divergence rate, shown in blue. Finally, using the 850 mb isosurface velocity field, the true trajectory divergence rate at the center point of the circle/flight radius was calculated, shown in red. Pearson correlation coefficients for these measurements can be found in Table 1.

Figure 7 shows the results for the attraction rate. As before, using measurements along perfectly circular paths, with radii fixed between 2 km and 15 km, from a 2D velocity field restricted to the 850 mb isosurface, the attraction rate was approximated, shown in black. Then, the attraction rate was approximated using velocity measurements from 3D simulated UAS flight paths, with radii fixed between 2 km and 15 km, attempting to follow the 850 mb isosurface, shown in blue. Finally, using the 850 mb isosurface velocity field, the true attraction rate at the center point of the circle/flight radius was calculated, shown in red. Pearson correlation coefficients for these measurements can be found in Table 2.

### 3.2. Using Eulerian Metrics to Infer Lagrangian Dynamics

In this section, results which indicate how well the attraction rate, $s_{1}$, and the trajectory divergence rate, $\dot{\rho }$, predicts Lagrangian dynamics, such as the passage of LCSs, are presented. Figure 8 shows the time series for the trajectory divergence rate and backward-time FTLE for integration times of 0.5, 1, and 2 h. Figure 9 shows the time series for the attraction rate and backward-time FTLE for integration times of 0.5, 1, and 2 h. The FTLE values have been multiplied by −1 for improved visualization. These values were calculated using velocity data from the 850 mb isosurface over the Kentland Farm portion of the NAM model.

The effectiveness of the attraction rate and the trajectory divergence rate for detecting LCSs was further explored using receiver operating characteristic (ROC) curves. ROC curves plots the true positive rate against the false positive rate for different threshold levels. Figure 10 shows an example of a true positive and a false positive from this study. To define these curves, we determined when LCSs passed within a threshold radius which ranged from 400 m to 10 km of the center point, Figure 11. We further applied a threshold of 90% for the LCSs, so only LCSs whose FTLE value was above the 90th percentile were considered. The attraction rate’s and the trajectory divergence rate’s ability to detect LCSs for integration times of 0.5,1, and 2 h in backward-time was explored. Figure 12 shows an idealized ROC curve with different threshold values; the farther the ROC curve is from the dotted line, the better the sensor is. This can be quantified using the area under the curve (AUC). The larger the AUC is above 0.5, the better the sensor is; a perfect sensor would have an AUC of 1.

Figure 13 shows ROC curves for the the trajectory divergence rate as measured from the center point of the sampling area (Figure 11). The threshold ranges from 0% at the upper right hand corner to 100% at the lower left hand corner. Every 20th percentile is marked with a dot. Each subplot represents a different threshold radius, with radii ranging from 400 m to 10 km. Each color represents a different integration time for the LCSs, 0.5 h green, 1 h red, 2 h blue. The AUC ranges from 0.477 to 0.756.

Figure 14 shows ROC curves for the attraction rate as measured from the center point of the sampling area (Figure 11). As before, the threshold is applied to the attraction rate ranging from 0%, upper right hand corner, to 100%, lower left hand corner. Every 20th percentile is marked with a dot. Each subplot represents a different threshold radius, with radii ranging from 400 m to 10 km. Each color represents a different integration time for the LCSs, 0.5 h green, 1 h red, 2 h blue. The AUC ranges from 0.626 to 0.850.

### 3.3. Inferring Lagrangian Dynamics from UAS Measurements

This section presents results which indicate how well the attraction rate, $s_{1}$, and the trajectory divergence rate, $\dot{\rho }$, as approximated from a simulated UAS flight, predict Lagrangian dynamics, such as the passage of LCSs, Figure 11. Figure 15 shows ROC curves for the the trajectory divergence rate as calculated from a simulated 2 km radius UAS flight. The threshold ranges from 0% at the upper right hand corner to 100% at the lower left hand corner. Every 20th percentile is marked with a dot. Each subplot represents a different threshold radius, with radii ranging from 400 m to 10 km. Each color represents a different integration time for the LCSs, 0.5 h green, 1 h red, 2 h blue. The AUC ranges from 0.491 to 0.742.

Figure 16 shows ROC curves for the attraction rate as calculated from a simulated 2 km UAS flight. As before, the threshold is applied to the attraction rate ranging from 0%, upper right hand corner, to 100%, lower left hand corner. Every 20th percentile is marked with a dot. Each subplot represents a different threshold radius, with radii ranging from 400 m to 10 km. Each color represents a different integration time for the LCSs, 0.5 h green, 1 h red, 2 h blue. The AUC ranges from 0.602 to 0.874.

## 4. Discussion

Looking at the results in Figure 6, we can see that the simulated UAS flight in 3D space provides a very similar result to the circular path restricted to the 850 mb isosurface. For all the radii that were analyzed, the trajectory divergence rate from the flight simulation is nearly identical to that from the idealized 2D circular path. Most of the error between the center point trajectory divergence rate and the estimate from the 3D flights appears to be due to the distance from the point of estimation, rather than inconsistencies in the flight’s path due to the UAS being buffeted by wind. This can also be seen in Table 1, where the correlation coefficients between the simulated flight and the 2D circle are all >0.95, while there is a steady drop in the correlation coefficients with the center point trajectory divergence rate as the radius increases.

Looking at the results in Figure 7, we see that the simulated UAS flight in 3D space provides very similar attraction rate measurements to the circular path restricted to the 850 mb isosurface. As before, for all the radii paths that were examined, the attraction rate from the flight simulation is nearly identical to that from the 2D circular path. Most of the error between the center point attraction rate and the estimate from the 3D flights appears to be due to the distance from the point of estimation, rather than inconsistencies in the flight’s path due to the UAS being buffeted by wind. This can also be seen in Table 2, where the correlation coefficients between the simulated flight and the 2D circle are all >0.96, while there is a steep drop in the correlation coefficients with the center point attraction rate as the radius increases.

Both the attraction rate and the trajectory divergence rate at a point can be approximated to a high degree of accuracy by UAS flights. Simulated 3D UAS flights provided measurements which were nearly identical to those of perfect circular 2D paths. The main cause of error in the approximations appears to be the radius of the circular arc. Furthermore, the trajectory divergence rate appears to be a more robust metric than the attraction rate, meaning that the trajectory divergence rate can be better approximated at larger radii than the attraction rate can. This can be seen very clearly in Table 1 and Table 2, where the correlation coefficient for the attraction rate drops off more quickly with flight radius than for the trajectory divergence rate.

As mentioned previously, the smallest flight radius which was explored in this study was 2 km. This radius was chosen because under 2 km, the spatial inhomogeneity of the simulated velocity field was insufficient to compute meaningful gradients. However, under current federal regulations in FAA part 107, unaided visual contact must be maintained with the UAS [32]. For smaller UASs, it is unlikely that an operator would be able to satisfy this requirement at a flight radius of 2 km. Fortunately, these results suggest that as the flight radius is decreased, the UAS approximation of the Eulerian diagnostic will converge the true value at the center point. Thus, we anticipate that in real-world experiments, where the flight radius will be on the order of hundreds of meters, a single fixed-wing UAS will provide an accurate approximation to the Eulerian diagnostics at the center point.

Figure 8 shows that the trajectory divergence rate does not always follow the trend of the negative backward-time FTLE. This is to be expected, as the trajectory divergence rate gives information on both instantaneous attraction and repulsion, while the negative backward-time FTLE gives only a measure of attraction. The trajectory divergence rate does, however, agree with the negative backward-time FTLE during periods of significant (large) attraction. This behavior is of particular interest for the detection of LCSs. When calculating LCSs, there is often a multitude of weaker, less important LCSs. In order to filter out these less important structures and focus on important structures, one often needs to set a threshold value for the FTLE field. These dips in the the trajectory divergence rate, coinciding with the strongest dips in the negative backward-time FTLE, would therefore seem to be a likely indicator of the most influential LCSs.

Figure 9 shows that the attraction rate follows the general trend of the negative backward-time FTLE. This is consistent as both the attraction rate and the negative backward-time FTLE give measures of attraction. The attraction rate appears to give a good approximation to the negative backward-time FTLE, and thus should be able to give indications of LCSs.

The ROC curves in Figure 13 show that the trajectory divergence rate, calculated at a point, can be used to detect the passage of LCSs. For the smaller threshold radii, AUC values for the ROC curves greater than 0.6 are consistently seen. This means that this method is outperforming random guessing, which would have an AUC of 0.5. Of course, as the threshold radius increases, the AUC trends to 0.5. This convergence to random chance at larger thresholds is consistent, since as the sample area increases, the likelihood of an LCS being within that domain will converge to 100%, at least for realistic atmospheric flows.

The ROC curves in Figure 14 show that the attraction rate, calculated at a point, cannot only be used to detect the passage of LCSs, but that it performs better at LCS detection than the trajectory divergence rate does. The smaller and more moderate threshold radii consistently display AUC values for the ROC curves of greater than 0.7, and many well over 0.8. This means that this method is far outperforming random guessing. Once again, it can be seen that as the threshold radius increases, the AUC trends towards 0.5, although this convergence is happening slower than it did with the trajectory divergence rate.

It should be noted that both the attraction and trajectory divergence rates seem to perform best at a threshold radius of around 0.8–2.0 km and become noisier as the radius decreases. We suspect that this is due to the spatial and temporal scales of the input data, 3 km × 1 h grid spacing. We speculate that with a velocity field that is either analytically defined or more highly resolved in both space and time, continued improvement in the ROC curves as the threshold radius decreases would be seen. Unfortunately, the analytical models currently used in the study of LCSs in 2D, such as the double gyre [34] and the Bickley jet [35], do not have the requisite spatial inhomogeneity necessary to reveal meaningful Eulerian structures in the attraction rate or the trajectory divergence rate fields.

The ROC curves in Figure 15 show that the trajectory divergence rate, as calculated from a simulated 2 km UAS flight, can be used to detect the passage of LCSs. For the smaller, and even more moderate, threshold radii, AUC values for the ROC curves greater than 0.6 are consistently seen. This means that this method is outperforming random guessing. Of course, as the threshold radius increases, the AUC trends to 0.5.

The ROC curves in Figure 16 show that the attraction rate, as calculated from a simulated 2 km UAS flight, cannot only be used to detect the passage of LCSs but once again performs better at LCS detection than the trajectory divergence rate does. For the smaller and more moderate threshold, radii consistently display AUC values for the ROC curves greater than 0.7, and many well over 0.8. This means that this method is outperforming random guessing. Once again, as the threshold radius increases, the AUC trends towards 0.5, although this convergence is happening slower than it did with the trajectory divergence rate.

In this paper, a planar circular flight path was used to determine if it is possible to detect an LCS passing through a given domain. However, this is just one potential application. More complicated flight paths could, hypothetically, be used to determine the size and shape of an LCS. For example, a corkscrew trajectory could potentially be used to determine the height of an LCS; likewise, a spiraling trajectory could potentially be used to determine the length of an LCS, or even to track an LCS as it moves.

The UAS-related results presented above were all calculated using Algorithm 1. Algorithm 1 assumes that the input scalar field is accurately measured and changing smoothly in time. In reality, experimental wind measurements are expected to have non-negligible error, both noise and bias types. It is still an open question as to how sensor error will affect these results. Future work will explore to what extent this LCS detection methodology is robust to that error.

## 5. Conclusions

We have put forward a novel algorithm to approximate the gradient of a scalar field using measurements from a circular arc around a point. Using realistic atmospheric velocity data from the NAM 3 km model, this algorithm was applied to circular trajectories restricted to a 2D isosurface and simulated UAS flights in 3D, with radii ranging from 2 km to 15 km. From these results, the trajectory divergence rate and the attraction rate were approximated for the center point of these paths. Comparing these approximations with the true trajectory divergence rate and attraction rate at the center point, we found that both the realistic flight simulator and the circle give nearly identical approximations. Furthermore, the approximations were very good for the smaller radii that were looked at, but even the larger radii approximations were able to pick up the trend of the trajectory divergence rate and attraction rate, though they underestimated the magnitude.

We have also examined the ability of Eulerian diagnostics, in particular the trajectory divergence and attraction rates, to infer Lagrangian dynamics. Using ROC curves, the ability of the trajectory divergence rate and attraction rate, as calculated at a point, to detect the passage of LCSs within a threshold radius was explored. We found that the attraction rate can be used as an effective tool to detect short term LCSs passing by. We also found that the trajectory divergence rate, while performing better than chance, underperformed the attraction rate. This analysis was then extended to look at the trajectory divergence rate and attraction rate as approximated by a UAS flight. Once again, we found that these Eulerian diagnostics, as approximated by a UAS flight, can be an effective tool for detecting LCSs passing through a sampling area.

This paper serves as a first step towards in situ detection of LCSs in the atmosphere. It demonstrates that a single fixed-wing UAS can, in principle, be used to measure Eulerian diagnostics of a local atmospheric flow. These Eulerian diagnostics can then be used to infer the Lagrangian dynamics of the local flow. Future work will apply this paper’s techniques to experimental data to detect real-world atmospheric LCSs using a fixed-wing UAS, evaluate the effects of sensor uncertainty on the accuracy of LCS detection, examine additional flight paths to attempt to determine size and shape of LCS, and extend the analysis to the detection of pollutant specific LCSs [5,12,36].

## Figures and Tables

**Figure 1 sensors-19-01607-f001:**
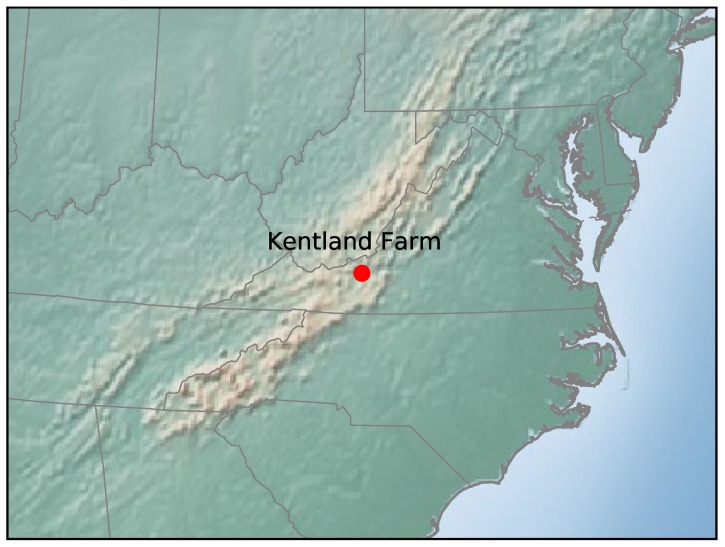
Map of the Mid-Atlantic United States, Kentland Farm in Virginia is shown in red.

**Figure 2 sensors-19-01607-f002:**
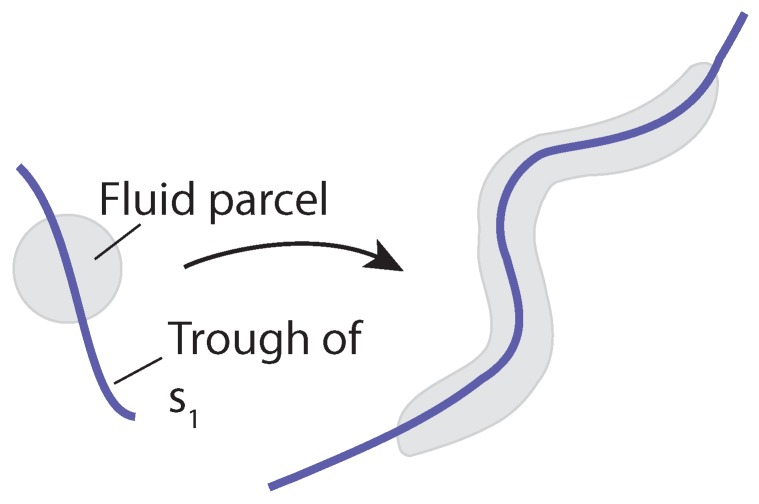
Schematic of the effect of a trough of the attraction rate, $s_{1}$, field on a fluid parcel.

**Figure 3 sensors-19-01607-f003:**
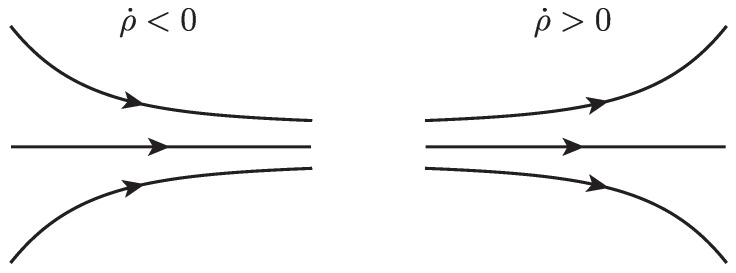
Schematic of the trajectory divergence rate: Where $\dot{\rho }<0$, trajectories are instantaneously converging; where $\dot{\rho }>0$ trajectories are instantaneously diverging.

**Figure 4 sensors-19-01607-f004:**
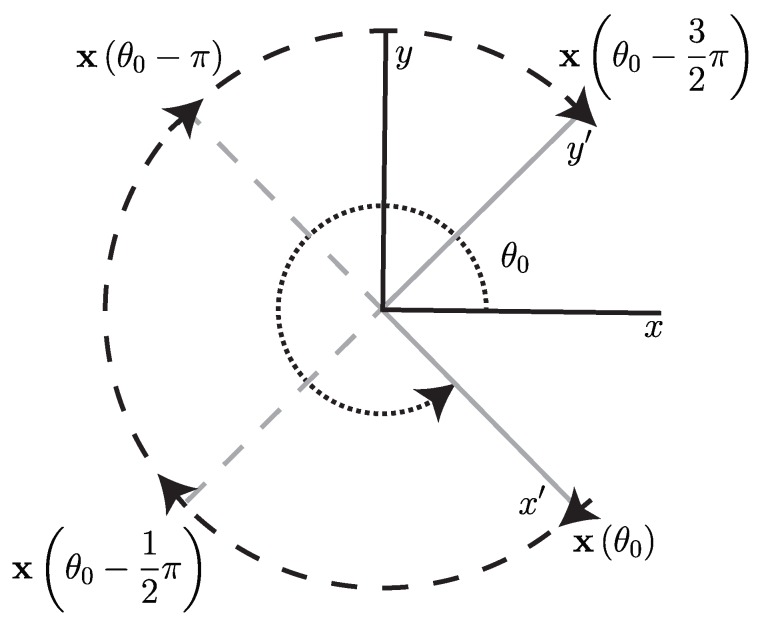
Schematic showing positions where velocity measurements were made and the position of the circle gradient frame to the reference frame.

**Figure 5 sensors-19-01607-f005:**
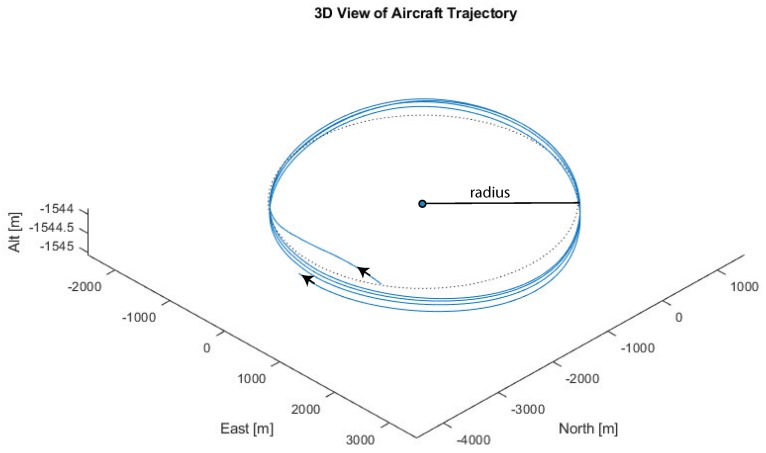
Simulated 3D unmanned aircraft systems (UAS) flight path (solid blue line) along with a 2D perfect circle (dotted black line). The 2D circle is at a constant altitude, while the 3D UAS flight tracks the 850 mb isosurface. The vertical dimension is shown highly exaggerated.

**Figure 6 sensors-19-01607-f006:**
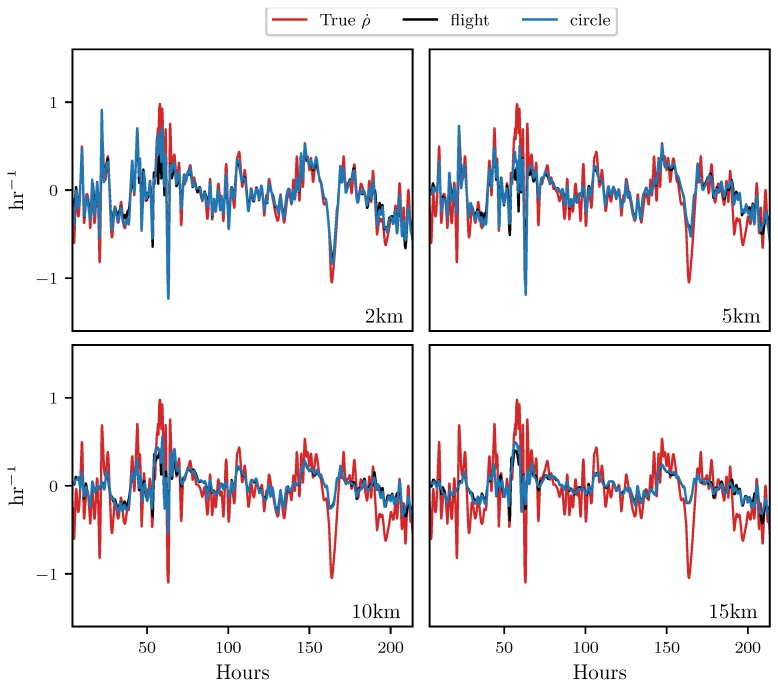
Comparison of measurements of the trajectory divergence rate $\dot{\rho }$ at the center point of the circular sampling orbit (**red**), along the path of a simulated UAS flight (**black**), and along a circular arc (**blue**). The radius of the circle and flight path is shown in the lower right hand corner of each subplot. The circular arc and the simulated UAS flight are nearly on top of each other.

**Figure 7 sensors-19-01607-f007:**
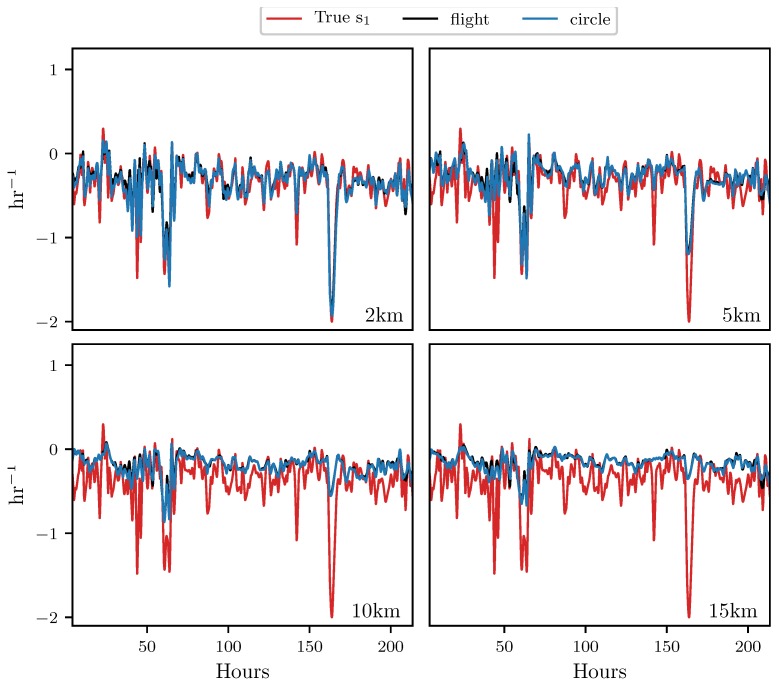
Comparison of measurements of the attraction rate $s_{1}$ at the center point of the circular sampling orbit (**red**), along the path of a simulated UAS flight (**black**), and along a circular arc (**blue**). The radius of the circle and flight path is shown in the lower right hand corner of each subplot. The circular arc and the simulated UAS flight are nearly on top of each other.

**Figure 8 sensors-19-01607-f008:**
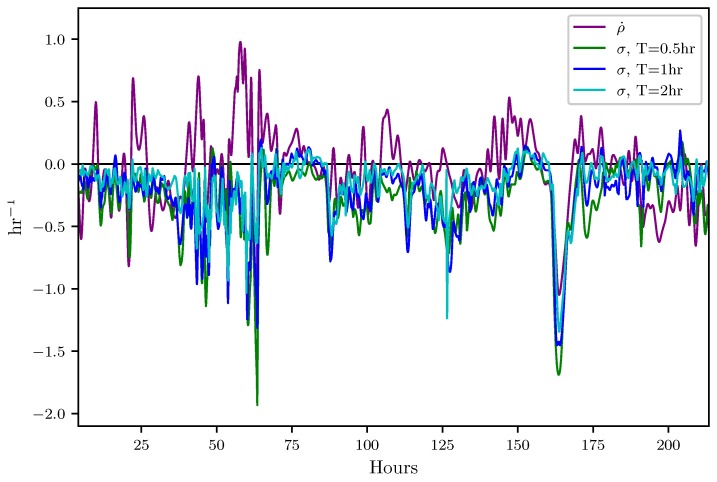
Comparison of the trajectory divergence rate with the 0.5,1, and 2 h backward-time finite-time Lyapunov exponent (FTLE) from t=4 to t=215 h. FTLE fields have been multiplied by −1 to offer better comparison of attraction.

**Figure 9 sensors-19-01607-f009:**
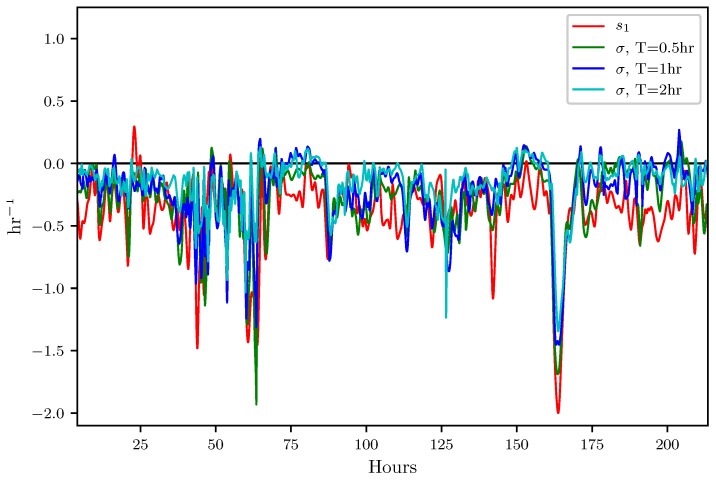
Comparison of the attraction rate with the 0.5,1, and 2 h backward-time FTLE from t=4 to t=215 h. FTLE fields have been multiplied by −1 to offer better comparison of attraction.

**Figure 10 sensors-19-01607-f010:**
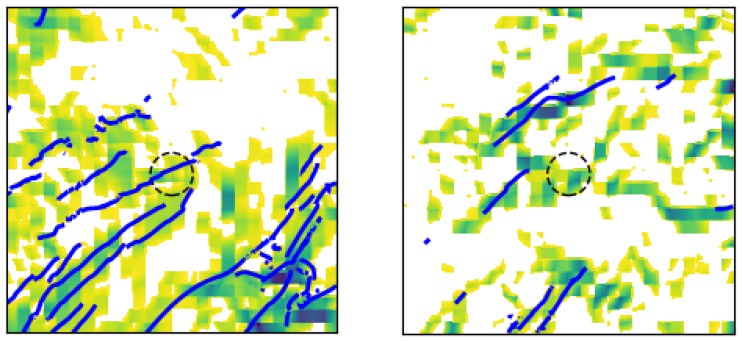
Depiction of a true positive (**left**) and a false positive (**right**), from approximately 43.5 h and 20.67 h, respectively. The true attraction rate field over Kentland Farm Va. is displayed; darker greens are lower values of the attraction rate. Grid points where the value of the attraction rate is above the threshold value are masked (white). Attracting LCS with an integration time of −1 h are shown as blue lines. The threshold radius is shown by the black dashed circle and has a radius of 5 km. In both examples, the attraction rate at the center of the sampling domain is below the threshold value, thus meeting the criterion for a positive identification of an LCS passing within the threshold radius. In the true positive example, there is an LCS passing within the threshold radius. Meanwhile, in the false positive, there is no LCS within the threshold radius. The ROC plot point for the particular case shown in this figure (i.e., threshold value and 5.0 km radius) is depicted by a red “+" marker in Figure 14.

**Figure 11 sensors-19-01607-f011:**
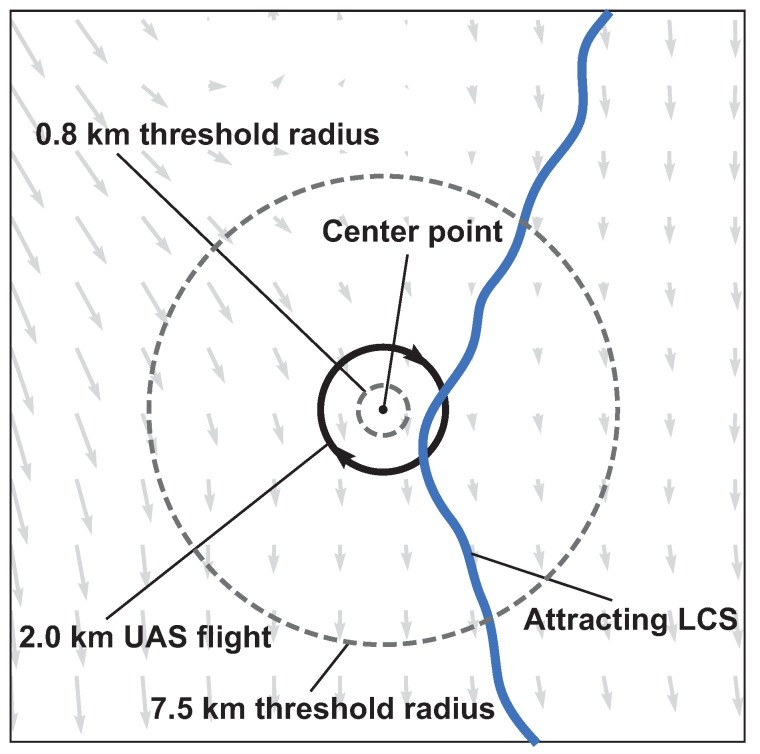
Schematic of Lagrangian coherent structure (LCS) detection showing two examples of threshold radii as dashed lines. An attracting LCS falls within the larger threshold radius but does not fall within the smaller radius. The instantaneous wind field is depicted as the background vector field.

**Figure 12 sensors-19-01607-f012:**
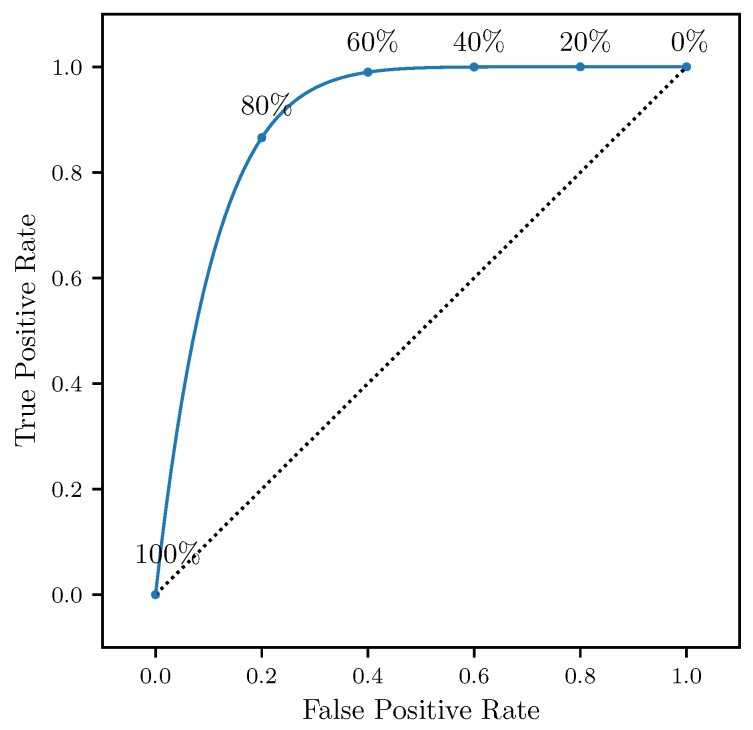
Schematic of an idealized receiver operating characteristic (ROC) curve and with threshold percentiles.

**Figure 13 sensors-19-01607-f013:**
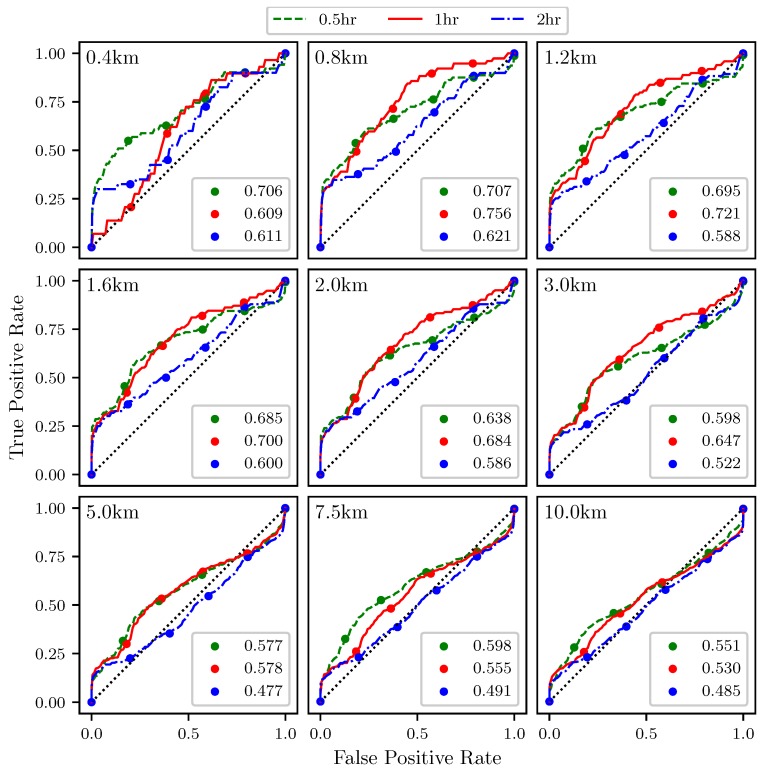
ROC curves for the trajectory divergence rate, $\dot{\rho }$, as measured at the center point ability to detect 90th percentile LCSs with integration times of 0.5 (**green**), 1 (**red**), and 2 (**blue**) h. Threshold radius for each subplot is displayed in the upper left hand corner. The area under the curve (AUC) for each integration time is given in the legend.

**Figure 14 sensors-19-01607-f014:**
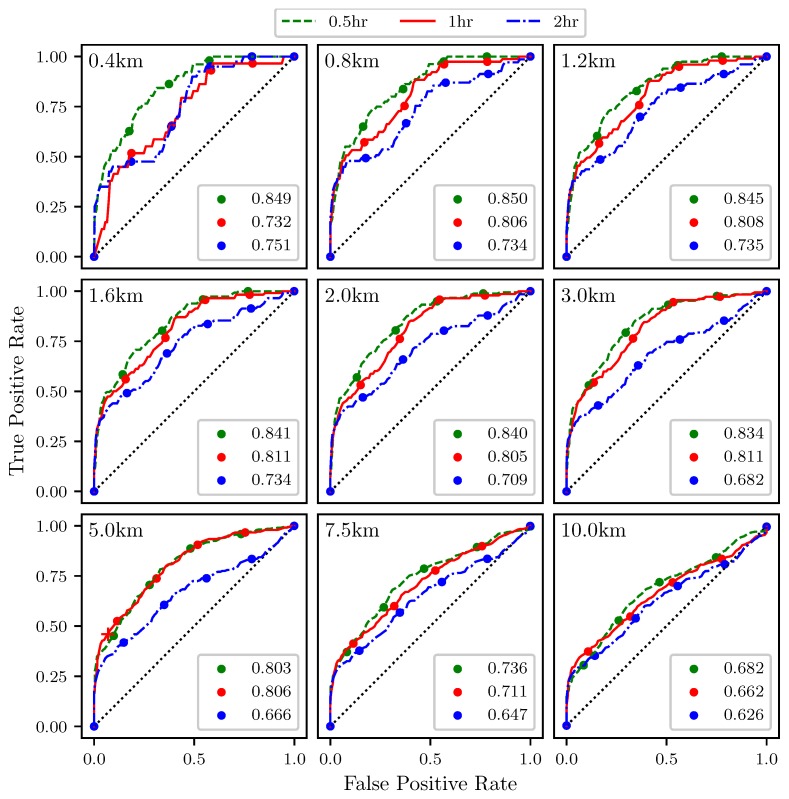
ROC curves for the attraction rate, $s_{1}$, as measured at the center point ability to detect 90th percentile LCSs with integration times of 0.5 (**green**), 1 (**red**), and 2 (**blue**) h. The threshold radius for each subplot is displayed in the upper left hand corner. The AUC for each integration time is given in the legend. The red “+” marker corresponds to the plot point for the case (i.e., threshold value and radius) that was shown in Figure 10.

**Figure 15 sensors-19-01607-f015:**
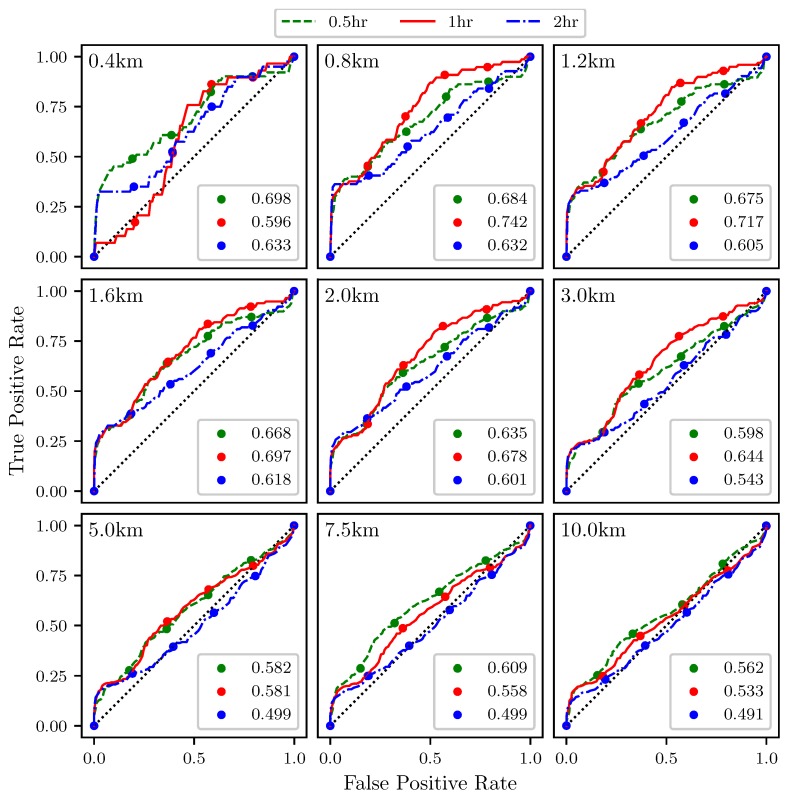
ROC curves for the trajectory divergence rate, $\dot{\rho }$, as measured from a 2 km radius UAS simulation ability to detect 90th percentile LCSs with integration times of 0.5 (**green**), 1 (**red**), and 2 (**blue**) h. The threshold radius for each subplot is displayed in the upper left hand corner. The AUC for each integration time is given in the legend.

**Figure 16 sensors-19-01607-f016:**
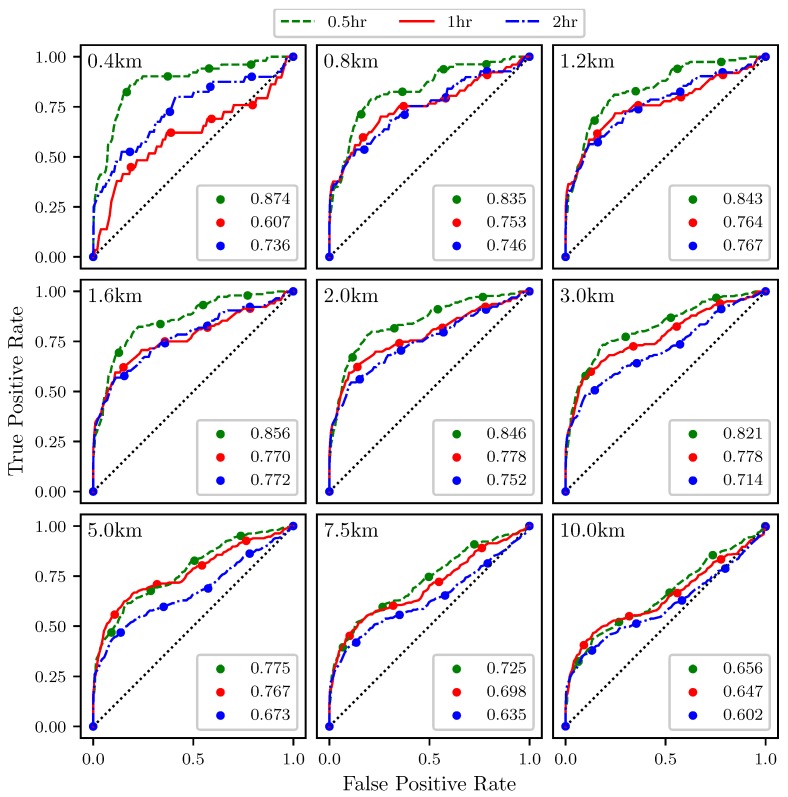
ROC curves for the attraction rate, $s_{1}$, as measured from a 2 km radius UAS simulation ability to detect 90th percentile LCSs with integration times of 0.5 (**green**), 1 (**red**), and 2 (**blue**) h. The threshold radius for each subplot is displayed in the upper left hand corner. The AUC for each integration time is given in the legend.

**Table 1 sensors-19-01607-t001:** Pearson correlation coefficients for trajectory divergence rate, $\dot{\rho }$, measurements. Coefficients range from 0.730 to 0.965.

	2 km Circle	5 km Circle	10 km Circle	15 km Circle	2 km Flight	5 km Flight	10 km Flight	15 km Flight
**center point**	0.955	0.854	0.790	0.730	0.931	0.827	0.781	0.730
**2 km circle**	--	0.946	0.815	0.751	0.981	0.923	0.811	0.765
**5 km circle**		--	0.866	0.768	0.935	0.981	0.865	0.784
**10 km circle**			--	0.928	0.804	0.836	0.974	0.902
**15 km circle**				--	0.745	0.738	0.904	0.955
**2 km flight**					--	0.944	0.824	0.783
**5 km flight**						--	0.870	0.793
**10 km flight**							--	0.937

**Table 2 sensors-19-01607-t002:** Pearson correlation coefficients for attraction rate, $s_{1}$, measurements. Coefficients range from 0.577 to 0.939.

	2 km Circle	5 km Circle	10 km Circle	15 km Circle	2 km Flight	5 km Flight	10 km Flight	15 km Flight
**center point**	0.939	0.838	0.677	0.590	0.910	0.821	0.675	0.577
**2 km circle**	--	0.932	0.742	0.644	0.980	0.917	0.739	0.627
**5 km circle**		--	0.898	0.789	0.916	0.980	0.887	0.760
**10 km circle**			--	0.908	0.729	0.881	0.978	0.864
**15 km circle**				--	0.637	0.788	0.907	0.965
**2 km flight**					--	0.936	0.746	0.644
**5 km flight**						--	0.900	0.791
**10 km flight**							--	0.909

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
