# Peer review of "A Method for Detecting Atmospheric Lagrangian Coherent Structures Using a Single Fixed-Wing Unmanned Aircraft System"

_sensors, 2019, doi:10.3390/s19071607_

Round 1

Reviewer 1 Report

The overall idea is good. My main suggestion is on the UAS wind part. 

(1) It is unclear what kind of UAS measurements are needed in algorithm 1. Maybe Va and Vgps vector? 

(2) It is also unclear how the sensing accuracy will affect the LCS estimation result. Algorithm 1 will be greatly affected if the sensing noise is untrivial because only four points are used. 

(3) The UAS is commanded to follow a circle as big as 2 km radius, which is kind of determined by the author's simulation platform. More discussions are needed here on how to design the flight trajectory based on LCS.

(4) Wind part is needed for Appendix. Also, are you using one set of CL/CD or not? 

Author Response

Please find our response in the attached pdf document.

Reviewer 2 Report

The paper address a very important issue related to new detection methodology of Lagrangian Coherent Structures (LCS).

The proposed method is much simpler than the ones described in recent literature.

The paper has a preliminary character as it is based on realist simulation, but the sound results herein point for a great potential of success in experimental realization.

The ROC curve study is a highlight of the paper and clearly shows the success of the proposed method.

The paper is very well written and free of mistakes.

Only a small correction on the caption of Fig 6 (Back instead of black) was detected. In my opinion, the paper can be published in its current form.

PS> I did not find necessary the appendix related to flight mechanics.

Author Response

(The authors gave the same response as above.)

Round 2

Reviewer 1 Report

The authors addressed most of my comments. I am fine with the revised version.